



# Lagrangian Gravity Wave spectra in the lower stratosphere of current (re)analyses

Aurélien Podglajen[1], Albert Hertzog[2], Riwal Plougonven[2], and Bernard Legras[2]

[1]Forschungszentrum Jülich (IEK-7: Stratosphere), Jülich, Germany
[2]Laboratoire de Météorologie Dynamique, Paris, France

*Correspondence to:* Aurélien Podglajen (a.podglajen@fz-juelich.de)

**Abstract.**

Due to their increasing spatial resolution, numerical weather prediction (NWP) models and the associated analyses resolve a growing fraction of the gravity wave (GW) spectrum. However, it is unclear how well this "resolved" part of the spectrum actually compares to the actual atmospheric variability. In particular, the Lagrangian variability, relevant, e.g., to atmospheric dispersion and to microphysical modeling in the Upper Troposphere-Lower Stratosphere (UTLS), has not yet been documented in recent products.

To address this shortcoming, this paper presents an assessment of the GW spectrum as a function of the intrinsic (air parcel following) frequency in recent (re)analyses (ERA-Interim, ERA5, the ECMWF operational analysis, MERRA-2 and JRA-55). Long-duration, quasi-Lagrangian balloon observations in the equatorial and Antarctic lower stratosphere are used as a reference for the atmospheric spectrum and compared to synthetic balloon observations along trajectories calculated using the wind and temperature fields of the reanalyses. Overall, the reanalyses represent realistic features of the spectrum, notably the spectral gap between planetary and gravity waves and a peak in horizontal kinetic energy associated with inertial waves near $f$ in the polar region. In the tropics, they represent the slope of the spectrum at low frequency. However, the variability is generally underestimated, even in the low-frequency portion of the spectrum. In particular, the near-inertial peak, although present in the reanalyses, has a much reduced magnitude compared to balloon observations. We compare the variability of temperature, momentum flux and vertical wind speed, which are related to low, mid and high frequency waves, respectively. The distributions (PDFs) have similar shapes, but show increasing disagreement with increasing intrinsic frequency. Since at those altitudes they are mainly caused by gravity waves, we also compare the geographic distribution of vertical wind fluctuations in the different products, which emphasizes the increase of both GW variance and intermittency with horizontal resolution. Finally, we quantify the fraction of resolved variability and its dependency on model resolution for the different variables. In all (re)analyses products, a significant part of the variability is still missing and should hence be parameterized, in particular at high intrinsic frequency. Among the two polar balloon datasets used, one was broadcast on the global telecommunication system for assimilation in analyses while the other is made of independent observations (unassimilated in the reanalyses). Comparing the Lagrangian spectra between the two campaigns shows that they are largely influenced by balloon data assimilation, which especially enhances the variance at low frequency.



# 1 Introduction

Atmospheric gravity waves (GWs) are mesoscale motions with large-scale impacts, notably through 3 mechanisms. First, they transport momentum from lower levels and deposit it higher up in the atmosphere, which forces large-scale circulations (Andrews et al., 1987), such as the Quasi-biennal oscillation (QBO, Baldwin et al., 2001). Second, they generate small-scale

turbulence (e.g., when breaking), which contributes to mixing atmospheric trace constituents (Podglajen et al., 2017) and diabatic heating. Third, GWs induce temperature and wind fluctuations which impact the formation and microphysical properties of clouds (e.g., cirrus clouds, Potter and Holton, 1995) and aerosols.

Because of those large-scale effects, GWs need to be represented in global atmospheric models. In current climate models with resolutions of the order of 100 km, GWs are mostly unresolved and need to be parameterized. On the contrary, global

weather forecast models, which currently have resolutions down to about 10 km or less, may now start to resolve a significant portion of the GW spectrum (e.g. Preusse et al., 2014; Jewtoukoff et al., 2015; Holt et al., 2016). However, the exact fraction of GWs resolved depends not only on the nominal resolution of the model, but also on the parameterized diffusion and on the representation of wave sources like tropospheric convection (Stephan et al., 2019). A common flaw of resolved GWs in models appears to be an underestimation of wave amplitude and an overestimation of horizontal wavelengths (e.g. Geller et al.,

2013; Holt et al., 2017). Even in models with realistic GW generation, a lack of realism in the propagation and dissipation of the waves often renders additional GW parameterization necessary in order to obtain a realistic general circulation (Holt et al., 2016, 2017). Hence, for a given model with a given resolution, it is not clear a priori what GWs are represented and what should be parameterized. This problem will become increasingly important as the models increase their resolution in the so-called "gray zone", resolving a larger part (but not all) the GW spectrum and its sources.

Crucial for climate and weather forecast models, the question of the fraction of resolved GWs is also important in (re)analyses. Those products have been used to investigate some properties of the GW field (e.g. Preusse et al., 2014). Reanalyses are also widely employed as input to trajectory calculations, notably (but not only) in the Upper Troposphere-Lower Stratosphere (UTLS), in order to understand, e.g., transport (e.g. Tzella and Legras, 2011), chemistry (e.g. Konopka et al., 2010) or cirrus cloud formation (e.g. Jensen and Pfister, 2004; Schoeberl et al., 2015; Ueyama et al., 2015). Among those three processes,

cloud formation (Spichtinger and Krämer, 2013; Dinh et al., 2016) is especially sensitive to mesoscale fluctuations of wind and temperature, so that a number of parameterizations have been developed to account for them (e.g. Bacmeister et al., 1999; Jensen and Pfister, 2004; Gary, 2006). A difficulty is that the parameterized fluctuations need to be similar to the perturbations experienced by air parcels (i.e., Lagrangian).

Recently, Podglajen et al. (2016b) used long-duration superpressure balloon (SPB) observations to characterize Lagrangian

wind and temperature fluctuations due to GWs in the lower stratosphere. SPB are especially suited for gravity wave studies since they follow the wind and directly provide access to the intrinsic frequency $\hat{\omega}$ of the waves. Podglajen et al. (2016b) proposed a parameterization of the vertical wind and temperature fluctuations for Lagrangian trajectory models that use (re)analyses to compute the trajectory path. A simpler-to-implement version of the parameterization approach and its salient features are presented in Kärcher and Podglajen (2019). However, both Podglajen et al. (2016b) and Kärcher and Podglajen





(2019), like other authors, implicitly assumed that most of the GWs seen in the observations were absent from the original reanalysis products. In the present paper, we compare GW induced fluctuations in modern (re)analyses from the same point of view as SPB, i.e. a Lagrangian point of view, to determine which part of the GW spectrum they resolve and how this resolved part compares with observations. Particular focus will be spent on the intermittency of the gravity wave fluctuations in the

(re)analyses.

Besides the interest for studies using Lagrangian trajectories, the comparison presented here also serves another purpose, this time from the point of view of model developers. For different models there are different criteria of success, a priori emphasizing the representation of the large-scale circulation. As the resolution of the models increases, a richer array of phenomena is present in the resolved fields and the question arises whether or not to assimilate information on these processes. In that

respect, assessing model errors on the GW field, which is described in both observations (e.g. radiosondes, GPS temperature profiles) and model output, is essential from the modelers' point of view. The long-duration balloon dataset provides a unique opportunity to perform such a task and assess the realism of modeled GW in the lower stratosphere.

The paper is organized as follows. Section 2 presents the balloon dataset and reanalyses used, as well as the comparison methodology. Then, the results and comparisons are presented in Sect. 3 and discussed in Sect. 4. Finally, summary and

conclusions are provided in Sect. 5.

## 2   Datasets and Methodology

### 2.1   Long-duration balloon observations

Most observations of the atmosphere are obtained either at a fixed location (Radar, lidar) or in motion relative to the flow (satellite, aircraft). In both cases, the relative speed between the measuring instruments and the air hampers direct Lagrangian

analysis of flow variability. One of the platforms which overcome that limitation are long-duration superpressure balloons (SPB), which provide observations in a quasi-Lagrangian frame of reference (e.g. Hertzog et al., 2002; Podglajen et al., 2016a). In the present study, we use SPB measurements as a reference to evaluate the representation of Lagrangian (quasi intrinsic-frequency) spectra. The observations were gathered in the Lower Stratosphere (16-20 km above sea level) during three SPB campaigns coordinated by the French Space Agency (CNES): Vorcore, PreConcordiasi and Concordiasi. Table 1 summarizes

the location and timing of the campaigns, as well as the sampling frequency and status regarding data assimilation. Vorcore and Concordiasi took place in the Southern polar vortex during austral spring, while Preconcordiasi flights were launched from an equatorial location (the Seychelles) in boreal spring. Whereas data from the later Concordiasi campaign was broadcast on the Global Telecommunication System (GTS), the earlier Vorcore and PreConcordiasi observations are independent datasets which can be used to evaluate reanalyses (Boccara et al., 2008; Podglajen et al., 2014). For that reason, we will focus in Sect. 3

on Vorcore and PreConcordiasi, while Concordiasi will be used to assess the impact of balloon data assimilation in Sect. 4.1. During the three campaigns, a whole set of measurements was performed, including e.g. ozone and particle measurements; for our purpose, however, only the horizontal position of the balloon from the on-board GPS, and pressure and temperature from the Thermodynamic SENsor (TSEN) will be used.



**Table 1.** Balloon measurement campaigns used as observational reference for this study. The last column (data assimilation) reports whether or not the data was broadcast on the Global Telecommunication System.

| Balloon campaign | geographic location | altitude (density) | number of balloons launched | period | measurement sampling | measurement precision [evtl. accuracy] | data assimilation |
|---|---|---|---|---|---|---|---|
| Vorcore | Southern polar vortex | 16-20 km (0.08-0.13 kg/m$^3$) | 25 | Sep. 2005 to Jan. 2006 | $u, v$: 15 min $p$: 1 min | $u, v$: $p$: | NO |
| Concordiasi | Southern polar vortex | 16.5-18 km (0.10-0.12 kg/m$^3$) | 19 | Sep. 2010 to Jan. 2011 | $u, v$: 1 min $p$: 30 s | $u, v$: $p$: | YES |
| PreConcordiasi | Tropics and southern mid-latitudes | 19-20 km (0.10-0.12 kg/m$^3$) | 3 | Feb. 2010 to May 2010 | $u, v$: 1 min $p$: 30 s | $u, v$: $p$: | YES |

Apart from occasional changes in the mass or volume of the system (such as dropsonde launching, changes in the incoming radiation flux) and balloon inertia acting at periods shorter than 15 min (Vincent and Hertzog, 2014; Podglajen et al., 2016b), SPB are essentially passively advected on isopycnic (constant-density) surfaces once they have ascended to their equilibrium level in the stratosphere. This quasi-Lagrangian behavior renders measurement interpretation relatively simple. Horizontal

winds $u$ and $v$ are deduced from the successive positions of the balloon, while pressure is directly measured by the TSEN meteorological sensors. In case of slightly uneven time sampling (varying sampling time step), the raw measurements were interpolated linearly onto a regular time grid. A fast Fourier transform algorithm was then applied to the time series, yielding the signal's Fourier transform $(\hat{u}(\hat{\omega}), \hat{v}(\hat{\omega}), \hat{p}(\hat{\omega}))$. Periodograms are then obtained directly from $|\hat{u}(\hat{\omega})|^2$, $|\hat{v}(\hat{\omega})|^2$ and $|\hat{p}(\hat{\omega})|^2$. In practice, we use a variant of Welch's method and estimate spectra by averaging periodograms obtained from 8-day windows

with 4 days overlaps. Note that for consistency with, e.g., Fritts and Alexander (2003), our definition of the intrinsic-frequency Fourier transform is such that the inverse transform reads:

$$(u(t), v(t), p(t)) = \int_{-\infty}^{+\infty} (\hat{u}(\hat{\omega}), \hat{v}(\hat{\omega}), \hat{p}(\hat{\omega})) e^{-i\hat{\omega}t} \mathrm{d}\hat{\omega} \tag{1}$$

While this sign convention does not affect the periodograms (estimated from squared modulus quantities and insensitive to the phase), it matters as far as the phase of the signal is concerned, notably for the polarization relations relating the Fourier

transforms of the different variables.

Combining the spectra of $|\hat{u}|^2(\hat{\omega})$ and $|\hat{v}|^2(\hat{\omega})$ leads to the horizontal kinetic energy spectrum $E_{k_h}(\hat{\omega})$ per unit mass:

$$E_{k_h}(\hat{\omega}) = \frac{1}{2} \left[ |\hat{u}|^2(\hat{\omega}) + |\hat{v}|^2(\hat{\omega}) \right] \tag{2}$$

On the other hand, pressure fluctuations $p'$ along the balloon trajectory can be used to estimate the vertical displacement of isopycnic surfaces $\zeta'_\rho$ and isentropic surfaces $\zeta'_\theta$ along air-parcel trajectories (see Hertzog et al., 2002; Podglajen et al., 2014,

2017):

$$\zeta'_\theta = \frac{1}{\alpha} \zeta'_\rho = -\frac{1}{g\bar{\rho}\alpha} p' \tag{3}$$





where $\overline{\rho}$ is the average segment density, $\alpha = \left(\frac{g}{C_p} + \frac{\mathrm{d}\overline{T}}{\mathrm{d}z}\right) \Big/ \left(\frac{g}{R} + \frac{\mathrm{d}\overline{T}}{\mathrm{d}z}\right)$ depends on the local temperature lapse rate $\mathrm{d}\overline{T}/\mathrm{d}z$ ($g$ is the gravitational acceleration, $C_p$ the thermal capacity of air at constant pressure and $R$ the gas constant for air). $\mathrm{d}\overline{T}/\mathrm{d}z$ is not directly measured but interpolated from the ECMWF ERA5 reanalysis; the sensitivity to the exact value of $\frac{\mathrm{d}\overline{T}}{\mathrm{d}z}$ is small and does not affect the conclusions presented below. Relation 3 relies on a few assumptions (small vertical excursions, small

Eulerian pressure perturbation relative to temperature perturbations, adiabatic and hydrostatic flow) which are well-met for the intermediate-frequency (periods between 1 day and 15 minutes) motions of interest here (as they are the ones resolvable by the reanalyses). Using Eq. 3, the potential energy spectrum per unit mass $E_p$ can be deduced from the estimated pressure spectrum:

$$E_p(\hat{\omega}) = \frac{1}{2}N^2|\zeta'_\theta|^2 = \frac{1}{2}\frac{N^2}{(g\overline{\rho}\alpha)^2}|\hat{p}|^2(\hat{\omega}) \tag{4}$$

As an illustration, Figure 1 (updated from Podglajen et al. (2016a)) shows the intrinsic-frequency spectra (power spectral densities) of $E_{k_h}$, $E_p$ and of the zonal momentum flux per unit mass $\overline{u'w'}$ for the Concordiasi (polar) and PreConcordiasi (tropical) campaigns, which have the highest sampling frequency (Table 1). The important characteristics that are present in the observations and will be searched for in the reanalyses are:

  - in the polar region, a spectral gap at $f$ between low frequencies and GW frequencies, seen in both $E_p$ and $E_{k_h}$. Then,
a local peak in $E_{k_h}$ at frequencies just higher than $f$, almost absent in $E_p$. In that frequency range ($f < \hat{\omega} < 4f$), $E_p < E_{k_h}$.

  - in the tropics and in the mid-frequency range ($\hat{\omega} \gg f$) for the polar latitudes, the spectra scale with a power-law behavior $\hat{\omega}^{-s}$ with $s \simeq 2$ for $E_p$ and $E_{k_h}$, and $s \simeq 1$ for $|u'w'|$. This scaling appears until $\hat{\omega} \simeq 100$ cy/day.

Above about 100 cy/day, the balloon-observed $E_p$ and $|u'w'|$ increase and peak near the Brunt-Väisälä frequency $N$. Al-
though there exist potential physical reasons for a spectral peak of momentum and potential energy near $N$ in the atmosphere (Podglajen et al., 2016a), part of the balloon-observed enhancement is likely an artifact caused by the non-isopynic response of the platform (Podglajen et al., 2016a; Vincent and Hertzog, 2014). Furthermore, due to their expected small horizontal scale the importance of non hydrostatic effects (absent from hydrostatic weather models), we can safely assume that buoyancy waves (near $N$) are absent from hydrostatic, low resolution analysis weather forecast models. Hence, we will focus our analysis to in-
trinsic frequencies smaller than 48 cy/day (periods larger than 30 min). For pressure observations, there is a significant aliasing at 48 cy/day from higher-frequency motions (in particular near the $N$-peak); this is overcome by averaging pressure observations at the highest-available frequency (1 min for Vorcore, 30 s for Concordiasi and PreConcordiasi, see Table 1) with a 15-minute Hann window weighting function and using this subsampled, low-pass-filtered version of the data in the subsequent analysis.

## 2.2 Atmospheric (re)analyses

For this study, we considered 4 recent reanalysis systems (ERA-Interim, ERA5, MERRA-2 and JRA55) and the ECMWF operational analysis (corresponding to the model versions used in February and December 2010, i.e. at the times of the PreConcor-





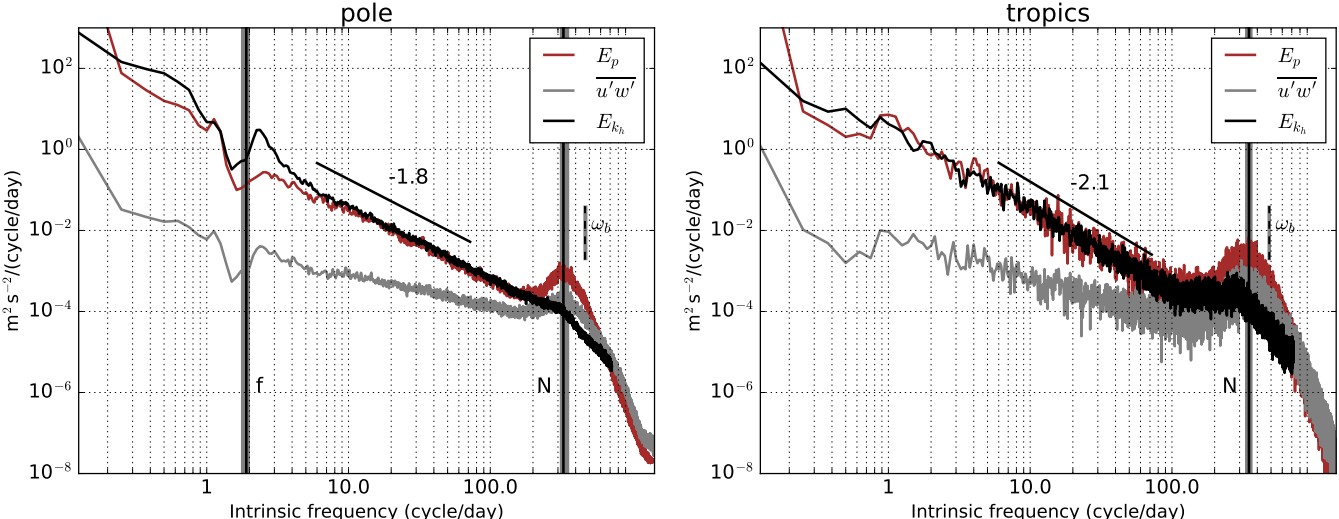

**Figure 1.** Average spectra of horizontal kinetic energy $E_{k_h}$, potential energy $E_p$ and $|\overline{u'w'}|$ inferred from SPB observations during the (left) polar and (right) equatorial balloon campaigns in 2010.

diasi and Concordiasi balloon campaigns). The model version and spatial resolution of the different products are summarized in Table 2, together with the time resolution at which the outputs (reanalyses and forecasts) are saved and/or made available. For more details regarding the model set-ups, physical parameterization and data assimilation flow, we refer the reader to the S-RIP introduction paper (Fujiwara et al., 2017).

Due to finite storage ability, the (output) time sampling is rather coarse (hourly at the best). This may have been a limiting factor for our study: GWs indeed have intrinsic periods ranging from 12 hours down to a few minutes (see Fig. 1), so that with fields output every 3 or 6 hours the dominant fraction of GW variance is aliased towards lower frequencies, thus potentially affecting our estimates. However, this limitation is mitigated by the fact that we analyse spectra as a function of intrinsic frequency $\hat{\omega}$:

$\hat{\omega} = \omega - k\bar{u} - l\bar{v}$                      (5)

combines the ground-based frequency of the motion $\omega$ and its horizontal scale (given by the zonal and meridional wavenumbers $k$ and $l$). Investigations of different time sampling with ERA5 (for which hourly outputs are available) demonstrated that, while in polar regions the considered intrinsic frequency spectra are strongly sensitive to a change of time sampling from 6 to 3 hours, even more frequent (than 3-hourly) sampling only marginally affected the results. In light of this, the main body of the

15 paper will not present the results obtained with JRA55, available only every 6 hours (see Table 2) for which our analysis suffers from aliasing. For a fair comparison, all (re)analyses (including ERA5) will be used at a time resolution of 3 hours, except in Appendix A where the impact of the output frequency is investigated.



**Table 2.** Description of the resolution of the models/(re)analyses used in this study. For the spectral models, N corresponds to the reduced Gaussian grid, F to the full Gaussian grid, and an approximate resolution is given. For the horizontal grid, cs: cubed sphere; sp: spectral model. Further information on the reanalysis systems can be found in Fujiwara et al. (2017).

| (Re)analysis | center | operational model version | horizontal grid type | Horizontal grid spacing | retrieved horizontal resolution | number of levels between 16 and 20 km | time frequency of the analysis (hours) | available time resolution with forecast (hours) | Reference |
|---|---|---|---|---|---|---|---|---|---|
| MERRA-2 | NASA | GEOS 5.12.4 (2015) | CS | $1/2°$ lat $\times 1/3°$ lon $\sim 60$ km | native | 4 | | 3 | Bosilovich et al. (2015) |
| JRA-55 | JMA | JMA GSM (2009) | sp | N160$\sim 55$ km | F160 | 3 | 6 | 6 | Kobayashi et al. (2015) |
| ERA-Interim | ECMWF | IFS Cy31r2 (2007) | sp | N128$\sim 79$ km | $(1°)^2$ | 4 | 6 | 3 | Dee et al. (2011) |
| ERA5 | ECMWF | IFS Cy41r2 (2016) | sp | N320$\sim 31$ km | $(1/8°)^2$ | 12 | 1 | 1 | |
| op. ECMWF | ECMWF | IFS Cy36r1 (2010) | sp | N640$\sim 16$ km | $(1/8°)^2$ | 8 | 6 | 3 | |

## 2.3 Comparison methodology

A simplistic approach to compare balloon observations and reanalyses would be to interpolate the model fields along the actual balloon trajectory. However, this might lead to erroneous conclusions regarding the variability present: if for instance the balloon were to record a vertical oscillation with a sheared flow $u$ in the reanalysis, this may lead to an oscillation in the

interpolated $u$ wind although they would be absent from an actual trajectory computed with the reanalysis wind. To avoid that complication, we directly computed isopycnic (balloon-like) trajectories using the reanalysis fields. In other words, we solve the following system of ordinary differential equations:

$$\begin{cases} \frac{\mathrm{d}X}{\mathrm{d}t} = u(X,Y,Z,t) \\ \frac{\mathrm{d}Y}{\mathrm{d}t} = v(X,Y,Z,t) \\ Z = \zeta_\rho(X,Y,Z,t) \end{cases} \tag{6}$$

the air density $\rho$ being a strictly decreasing function of geometric altitude. In practice, System 6 is solved with $\ln(\rho)$ as the

vertical coordinate, and the 2D trajectories are integrated using a Runge-Kutta scheme of order 4 and a time step of 1 minute, adjusted when needed to satisfy the CFL criterion. We note that the dependency of the trajectory on the integration time step and the details of the numerical scheme is small compared to other sources of errors (Bowman et al., 2013). Gridded reanalyses fields (typically $p$, $T$, $u$ and $v$) are interpolated in the horizontal and time dimension using cubic splines, leading to vertical profiles; then the vertical coordinate $\ln(\rho)$ is calculated and finally the wind is interpolated at the density level of the balloon. To

examine the Lagrangian variability and estimate spectra, we calculate 8-days trajectories started at the balloon position every 4 days, thus matching the segments used in the observations. Examples of such trajectories for ERA5 reanalysis are displayed in Fig. 2.

   At this point, two remarks should be made. First, regarding trajectory accuracy: while it is affected by the sampling frequency (in space and time) of the model and the interpolation method used (Stohl et al., 1995; Bowman et al., 2013), the main source

of uncertainty in the lower stratosphere stems from errors in the reanalysis fields (Boccara et al., 2008; Podglajen et al., 2014).





In polar regions, the analyses compare well to observations at lower stratospheric altitudes (Boccara et al., 2008), so that there is generally not much difference between observed and simulated balloon trajectories over periods of ~8 days, as illustrated in Fig. 2 for the case of ERA5. In the tropics, on the contrary, analyses may exhibit large deficiencies (Podglajen et al., 2014; Dharmalingam et al., 2019), and the computed trajectories can largely diverge from the observed ones. However, our results are
5    not overly sensitive to the exact number and location of trajectory segments used. Errors in the large-scale wind are deemed very unlikely to generate a sampling bias and artificially degrade GW variability (which they could do in particular and improbable cases, e.g., by making the balloons drift systematically in quieter areas than those where they actually flew).

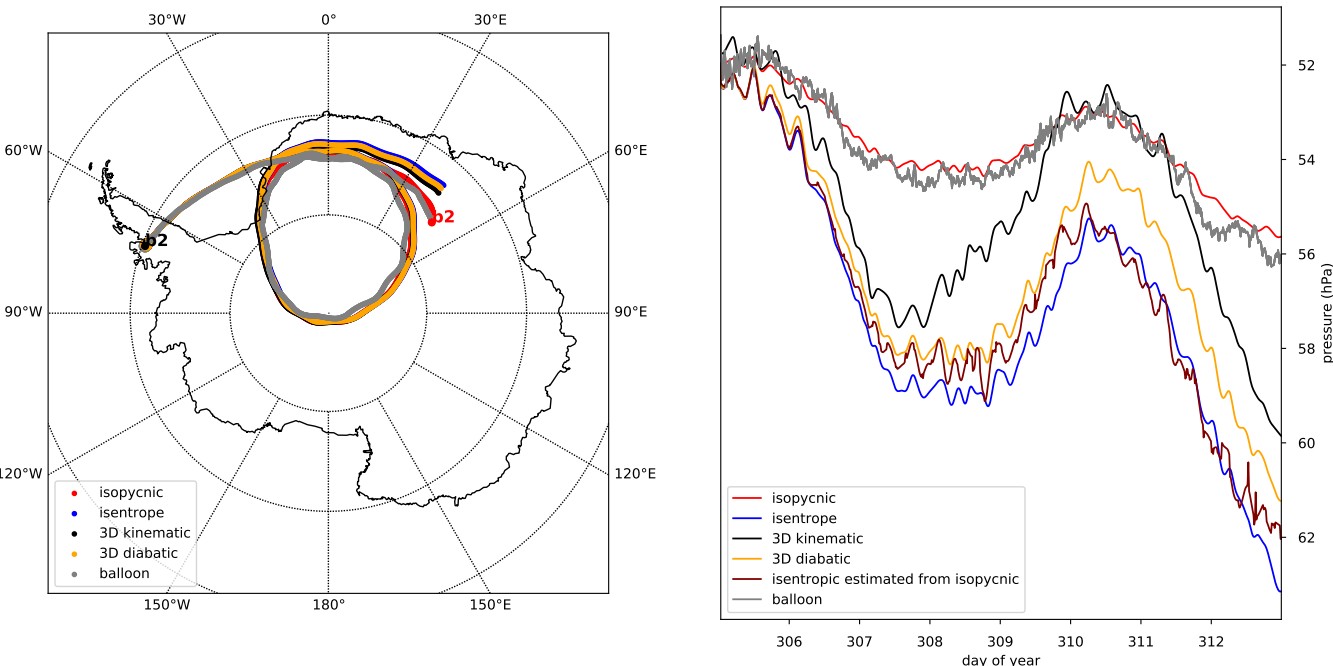

**Figure 2.** (Left) Observed and calculated horizontal trajectories of Vorcore Balloon 2 started from its position on 2005/11/02 at 00:00 UTC. The calculated trajectories are 3D-kinematic (with omega velocity, black), 3D-diabatic (with diabatic heating rates, orange), 2D-isentropic (blue) and 2D-isopycnic (red) trajectories; all were computed using ERA5 wind and temperature field at 3-hourly output resolution. The observed trajectory of the balloon is shown in grey. (Right) Pressure time series corresponding to the horizontal trajectories on the left panel.

Second, we note that physical applications and process studies are interested in air parcel trajectories, which are a priori distinct from the isopycnic trajectories of the balloons. However, as argued above and in Podglajen et al. (2016b), the horizon-
10    tal positions of isopycnic, isentropic and true air parcel trajectories should remain close (relative to the distance traveled) at synoptic time scales (a few days), and Equation 4 is expected to be a good approximation in the stratosphere with low diabatic heating. In order to verify this, we compared the 8-day trajectories and spectra for isopycnic, isentropic (constant potential temperature $\theta$) and full-3D trajectories, computed either in $p$ coordinate with $\omega = \mathrm{D}p/\mathrm{D}t$ as vertical velocity (kinematic tra-





jectories) or with $\theta$ coordinate and $\dot{\theta}$ vertical velocity (diabatic trajectories). The different horizontal and vertical trajectories for the special case of Vorcore balloon 2 on 2005/11/02 are displayed on Fig. 2.

Figure 2 demonstrates the validity of our approximations for ERA5: the 3 trajectory types only slightly diverge from one another in the horizontal. Although as expected the isopycnic trajectory diverges more rapidly from the 3D trajectory than the

isentropic one, for an integration time of 8 days the paths remain close. While their vertical positions are clearly distinct, the isentropic, 3D-kinematic and isopycnic (scaled onto the isentropes using Eq. 3) trajectories are highly correlated at short time scales. As examined in Appendix B, the estimated Lagrangian energy spectra exhibit only slight differences. Hence, for a more direct comparison to balloon observations, we considered isopycnic trajectories (adjusted using Eq. 3) in the following.

## 3 Results: Intrinsic frequency spectra of Lagrangian fluctuations

In Fig. 3 are depicted the intrinsic frequency spectra (power spectral density or PSD) of $E_{k_h}$, $E_p$ and the zonal pseudo-momentum flux $(1 - f^2/\hat{\omega}^2)|u'w'| = (1 - f^2/\hat{\omega}^2)\Re(\hat{u}\hat{w}^*)$ derived from isopycnic trajectories computed using 3-hourly wind data from the different (re)analyses. The trajectories are output every 15 minutes, so that from the point of view of the sampling, the highest resolvable frequency (Nyquist frequency) in the spectra is $1/48$ cy/day. However, we should recall that, as mentioned above and explained in Appendix A, the effective resolution in terms of intrinsic frequency actually depends the

limited time and the space resolution of the (re)analyses.

### 3.1 Horizontal kinetic energy spectra $E_{k_h}(\hat{\omega})$

The top two panels of Fig. 3 show the $E_{k_h}$ spectra (solid lines) obtained from polar (Vorcore, left) and equatorial (PreConcordiasi, right) trajectories for the balloon observations (black lines) and the reanalyses (colors). In the polar case, low frequency features (below the Coriolis frequency $f$) are well-represented all reanalyses. However, in the gravity wave frequency-range

(above $f$), the variance in $E_{k_h}$ is largely underestimated in all 3 systems compared to observations. Nevertheless, the reanalysis $E_{k_h}$-spectra exhibit typical features similar to the observations, including:

1. a local minimum in $E_{k_h}$ between $f/2$ and $f$, the characteristic spectral gap separating GW ($\hat{\omega} > f$) from synoptic-scale motions ($\hat{\omega} < f$)

2. a local maximum in $E_{k_h}$ around $1.2 - 1.5\ f$ (the near inertial peak, see Hertzog et al. (2002))

3. a power-law decrease in $E_{k_h}$ for frequencies larger than $1.5\ f$

The differences between reanalyses and observations mainly concern:

1. the frequency at which the peak around $f$ in $E_{k_h}(\hat{\omega})$ occurs and its magnitude.

2. the power-law slope of the decreasing $E_{k_h}$ for $\hat{\omega} \gg f$. The observations have the shallower slope, followed by ERA5 and ERA-Interim, MERRA-2 having the steepest slope.





Despite the quantitative disagreement between the different reanalyses, they also exhibit qualitative similarities with one another and with the observations. In particular, they all show a near-inertial $E_{k_h}$-peak, consistent with observations (Hertzog et al., 2002).

In order to further investigate the agreement between the resolved perturbations and gravity wave theory, we examine to
what extent the polarisation relation for inertio gravity waves are fulfilled. On the $f$-plane, with the convention given by Eq. 1, the polarization relation for the horizontal wind reads (Fritts and Alexander, 2003):

$$\hat{u}_\parallel(\hat{\omega}) = i\frac{\hat{\omega}}{f}\hat{u}_\perp(\hat{\omega}) \tag{7}$$

where $\hat{u}_\parallel$ is the amplitude of the horizontal wind along the wave vector and $\hat{u}_\perp$ the amplitude perpendicular to the wave vector. Keeping in mind the convention expressed by Eq. 1, Equation 7 indicates that low-frequency waves (near $f$) induce
anticyclonic flow rotation whereas high frequency waves have their horizontal wind perturbation aligned with the wave vector and no preferred rotation direction. To make use of this property and further demonstrate that the near-inertial peak in the analyses is due to inertial oscillations, we turn to rotary spectral analysis. The rotary Fourier transform of the horizontal wind, $\hat{U}$, is defined by:

$$\hat{U}(\hat{\omega}) = \hat{u}(\hat{\omega}) + i\hat{v}(\hat{\omega}) \tag{8}$$

As a consequence of Eq. 7, the rotary ratio $R(\hat{\omega})$, ratio of the PSD of anticlockwise horizontal motions ($|\hat{U}(-\hat{\omega})|^2$ with our Fourier transform convention in Eq. 1) and clockwise ones ($|\hat{U}(\hat{\omega})|^2$), verifies the relation:

$$R(\hat{\omega}) \equiv \left|\frac{\hat{U}(-\hat{\omega})}{\hat{U}(\hat{\omega})}\right|^2 = \left(\frac{1 - \frac{\hat{\omega}}{f}}{1 + \frac{\hat{\omega}}{f}}\right)^2 \tag{9}$$

Equation 9 has been exploited in previous studies (e.g. Hertzog et al., 2002; Conway et al., 2019) to evaluate the consistency of the observed horizontal wind spectrum with linear gravity wave theory. In particular, the dominance of anticyclonic (here
in the Southern hemisphere) motions at low frequencies is consistent with Eq. 9 and indicates the importance of linear gravity waves, opposed to, e.g. stratified turbulence for which no systematic phase relation between the horizontal wind components is expected. In Fig. 3, $R(\hat{\omega})$ is displayed together with its theoretical value ratio given by Eq. 9. The dominance of anticyclonic motions is a further argument in favor of inertio-GW being responsible for the $f$-peak in both reanalyses and observations. However, we note that it is less pronounced in the reanalyses than in the observations, and in ERA5 than ERA-Interim and
MERRA-2. Possible reasons for this will be examined in the next section.

In the tropics, the statistical agreement between reanalyses and observations regarding the representation of high-frequency variability is surprisingly better than in the polar latitudes. This contrasts with the representation of large-scale wind field which is better in polar regions (Boccara et al., 2008) than in equatorial ones (Podglajen et al., 2014). The agreement obtained for low-frequency waves is quantitative as well as qualitative, and reaches up to a cut-off frequency which is about 4 cycles/day for the
ERA-Interim, ERA5 and the ECMWF operational analysis, and 2 cycles/day for MERRA-2. Below this cut-off frequency, the observed and modeled age spectra are in quantitative as well as qualitative agreement. At the cut-off frequency, the reanalysis





spectra show a kink and above it they drop with much larger slopes than the observed ones, which keep decaying with a constant slope.

## 3.2 Potential energy spectra

The middle two panels of Fig. 3 show the spectra of the potential energy per unit mass $E_p$, for the polar (left) and tropical (right)
flights. To a large extent, the situation is similar to the one described above for $E_{k_h}$: in the polar case, there is a qualitative agreement in the structure of the power spectra (notably regarding the spectral gap around $f$), but a quantitative disagreement and a steeper power law-slope than in the observations.

Besides the direct value of the potential energy, it is again interesting to investigate whether the fluctuations in reanalyses verify the polarisation relation expected for gravity waves. In the absence of constructive interference, the ratio of potential to
horizontal kinetic energy for frequencies $\hat{\omega} \ll N$ should obey (Podglajen et al., 2016b):

$$\frac{E_{k_h}(\hat{\omega})}{E_p(\hat{\omega})} = \frac{\hat{\omega}^2 + f^2}{\hat{\omega}^2 - f^2} \tag{10}$$

Figure 3 exhibits $\frac{E_{k_h}(\hat{\omega})}{E_p(\hat{\omega})}$ for the 3 reanalyses and the observations (dashed lines), together with its theoretical value given in Eq. 10. The observations again closely follow theoretical expectations from $\sim 1.2\,f$, with a dominance of kinetic energy, to higher frequencies (the so-called mid-frequency range) where there is an equipartition between $E_p$ and $E_{k_h}$. The variability
in reanalysis trajectories shows the same evolution with frequency as the observations, and the $E_{k_h}/E_p$ ratio decreasing with increasing frequency. However, reanalysis also suffer from a significant overestimation of the kinetic energy compared to the potential energy. This is the case by a factor 1.5 to 3 for ERA5, and 5 for the ERA-Interim and MERRA-2. Together with the rotary ratio analysis, this suggests that a significant fraction of the variability does not obey the observed (and expected) polarisation relations for gravity waves.

In the tropics, as for the $E_{k_h}$ spectra, the $E_p$ spectra from reanalyses and observations are in better agreement than over the pole. In particular, the $E_{k_h}/E_p$ ratio is close to 1 for the reanalyses, as expected, whereas there is a quantitative agreement between observed and analyzed $E_p$ up to a threshold frequency similar to that encountered with $E_{k_h}$. Beside the reanalyses, for the tropical case the ECMWF operational analysis trajectories are also displayed, which shows similar quantitative results as for the ERA5 reanalysis.

## 25  3.3 EP flux and vertical wind spectra

The bottom two panels of Fig. 3 display the zonal pseudo-momentum $\left(1 - f^2/\hat{\omega}^2\right)|u'w'|$ flux spectra of resolved waves in the reanalyses. This quantity characterizes the forcing of the large-scale flow by the waves so that those panels enlighten us regarding the missing gravity wave drag from resolved waves compared to observations.

Over the pole, the reanalyses underestimate the variability in the whole GW range, with ERA5 being the closest to the
balloon observations. Although the number of reanalysis products is limited, there appears to be a correlation between the fraction of variability resolved at low frequency and the vertical resolution of the product (given in Table 2). With the highest





vertical resolution, ERA5 resolved the largest fraction of the variability, followed by ERA-Interim and MERRA-2 with coarser resolutions.

In the equatorial region, on the other hand, similar to the $E_{k_h}$ and $E_p$ spectra, the momentum flux by low-frequency equatorial GW is comparable to observations in all products, up to the threshold frequency where it drops with a slope twice as large

as the actual slope.

We do not show the vertical wind spectrum, but its shape can be readily deduced from the potential energy spectrum as $E_{k_v} = \frac{1}{2}\hat{w}^2 \propto \left(\frac{\hat{\omega}}{N}\right)^2 E_p$, so that the inequal performance of the reanalysis can be deduced from the middle panel of Fig. 3. It should be noticed that the different quantities considered have changing power-law slopes in the gravity wave range from $\hat{\omega}^{-2}$ for the horizontal kinetic energy $E_{k_h}$ and potential energy $E_p$ power spectra, to $\hat{\omega}^{-1}$ for the EP flux spectrum $\left(1 - f^2/\hat{\omega}^2\right)|u'w'|$,

and to $\hat{\omega}^{\sim 0}$ for the vertical kinetic energy $E_{k_v}$. As a consequence, the variability of different fields emphasizes different parts of the spectrum: while $u$, $v$ ($E_{k_h}$) and $T$ ($E_p$) are more connected to the low-frequency part of the gravity wave spectrum, $\left(1 - f^2/\hat{\omega}^2\right)|u'w'|$ corresponds to the intermediate frequencies, and the vertical wind component $w$ (related to $E_{k_v}$) to the high-frequency waves.

### 3.4   Intermittency and distribution of the fluctuations

The PSDs examined above inform on the autocorrelation and the repartition of the fluctuations as a function of intrinsic frequency. However, as pointed out by several authors (e.g. Hertzog et al., 2012; Podglajen et al., 2016b), the probability distributions are also relevant to determine whether the variance is due to ubiquitous, constant variability generated by the superposition of random waves, or if rare, large excursions created by specific wave events can occur. This last behavior is expected for GW which are known to be intermittent.

#### 3.4.1   Intermittency of the fluctuations

Similar to what was used for the Lagrangian spectra, we compare the distribution of the fluctuations in the reanalyses to the observations based on computed isopycnic trajectories corrected for the non-isopycnic behavior (by using the coefficient $\alpha$ defined in Eq. 3). In order to focus on the GW range, we filter the outputs of the trajectories to keep only the signal corresponding to intrinsic frequencies shorter than 48 cycles/day and larger than $f(30°)/(2\pi) \simeq 1$ cycle/day (tropics) or $f(70°)/(2\pi) \simeq 2$

cycles/day (pole). The lower bound is considered as the lower bound of the GW frequency range in the regions studied, while the upper bound corresponds to the Nyquist frequency of Vorcore balloon position information (every 15 minutes). It is below the frequency at which the balloons start to depart significantly from the isopycnic behavior, so that the comparison can only be marginally affected by the non-isopycnic balloon response. In that frequency range $[f/(2\pi); \sim (N/2)/(2\pi)]$, we consider temperature, momentum flux and vertical velocity to characterize statistics and intermittency of respectively low, medium and

high-frequency waves.

Figure 4 shows the probability distribution (PDF) of the three quantities (temperature, momentum flux and vertical wind) for all the (re)analyses considered. From top to bottom, the considered fluctuations are primarily induced by waves of increasing frequency; $|T'_l|^2 \propto E_p(\hat{\omega})$ emphasizes the low frequencies while $|u'w'| \propto \hat{\omega} E_p(\hat{\omega})$ and $|w'|^2 \propto \hat{\omega}^2 E_p(\hat{\omega})$ are related to increas-



ing frequencies. Since high frequency waves are more poorly represented than low frequency ones (Fig. 3), it comes with no surprise that the PDF of analyzed and observed fluctuations are in increasing disagreement from temperature to momentum flux and from momentum flux to vertical velocity in both tropical and Southern polar regions.

The width of the temperature PDFs for the polar and tropical regions is redundant with the level of the variance in the $E_p$

spectrum already shown in Fig. 3. The additional information provided in Fig. 4 concerns the shape of the PDF. In the tropics, GW temperature fluctuations are ubiquitous and characterized by a Gaussian PDF. This behavior is fairly well reproduced in all reanalyses examined here. In contrast, in the pole, the balloon PDFs are no longer Gaussian. In particular, they have longer tails than Gaussian PDFs, a behavior which is reproduced by reanalyses.

Regarding the momentum flux, the tropical and polar PDFs both exhibit log normal PDFs, as well as the reanalyses. We note

that the tropical regions during PreConcordiasi show more activity than the polar regions during Vorcore, but this is largely due to a wider GW range in the tropics, where $f \rightarrow 0$. Finally, the vertical wind PDFs show Laplace distributions in both cases, and the order of the reanalyses is the same as for the rest. In summary, while the fraction of resolved variance varies from one analysis to the other, it appears that the basic intermittency properties of the GW field and the shape of the PDF of the fluctuations are consistent between observations and the different reanalyses.

It should also be noticed that the conclusion regarding the resolved fraction of variability for the different reanalyses are transferable from one campaign to the other. In other terms, for all quantities and regions examined here, we find that the more realistic reanalysis is ERA5 followed by ERA-Interim and MERRA-2. The 2010 ECMWF operational model performs better than ERA5, likely because of its superior horizontal resolution.

### 3.4.2 Geographic distribution of the fluctuations

We have considered above the intermittency of GWs sampled by the balloons along their trajectories. This intermittency stems from both space and time variability of the GW field in which the balloons drift. A significant part of it comes from the geographic variability of wave sources, as illustrated in Fig. 5 through global maps of the vertical wind standard deviation at 100 hPa (related to high frequency wave activity) for the month of March 2010. The geographic structure is similar between the different reanalyses, which is consistent with the fact that intermittency is similar in the different products (see the shapes in

Fig. 4). GW amplitudes largely differ from one product to the other, but the geographic variability is to a large extent common to all reanalyses. In the season examined (boreal spring 2010), mountain ranges such as the Rockies, the Andes, the Himalaya or the Antarctic peninsula stand out as regions of large activities. Convective regions such as the Intertropical Convergence Zone are also characterized by a larger activity in all products. However, although the general geographic pattern of $\sigma_w$ do match between the different products, there are differences in the details. In particular, the features are sharper and smaller

scale (especially around orography) in the high resolution products (the operational analysis and ERA5) than in the lower resolution reanalyses. Although this property might be expected, it implies that intermittency is increased in those products.

Since the different resolutions in the reanalyses imply different horizontal wave numbers which may undergo different filtering and vertical propagation, we present a vertical profile of $\sigma_w$ in the tropics for the different products in Fig. 6, panel a). There is a drastic reduction in $\sigma_w$ from the troposphere to the stratosphere, related to the increased stratospheric stability



hampering vertical motions there. Fig. 6, panel b) displays the same $\sigma_w$ equatorial profile but normalized by the standard deviation from the ECMWF analysis. Although they do not resolve the same wave population, the different analyses essentially show similar vertical structure in GW variance, except ERA5 for which $\sigma_w$ is relatively reduced around the tropopause. This is likely due to the increased stability in that region in the high vertical resolution ERA5.

## 4  Discussion

### 4.1  Impact of balloon data assimilation

Balloon observations from the PreConcordiasi and Vorcore campaigns were not assimilated in the reanalyses, so that they provide an independent evaluation of the resolved GW variability. However, data collected during the Concordiasi campaign, which took place in austral winter 2010, was broadcast on the GTS and assimilated in most analyses, including ECMWF operational data, ERA5 and ERA-Interim, and MERRA-2 (Hoffmann et al., 2017). Hence, comparing the period of the (assimilated) Concordiasi campaign with that of the (unassimilated) Vorcore provides an opportunity to characterize the extent to which balloon data assimilation in the atmospheric models impacts the representation of the wave field (either through spurious wave generation or realistic waves introduced in the initial state that propagate in the forecast). Since the Concordiasi dataset was only present one year, GW activity in the 2010 southern lower stratospheric polar vortex is not typical but largely influenced by data assimilation. However, with the development of the Loon dataset of superpressure balloons (Conway et al., 2019) and studies regarding its assimilation in NWP models (Coy et al., 2019), it is interesting to document the impact of such a SPB dataset.

The $E_{k_h}$ spectra for the two campaigns are shown in Fig. 7. There are slight differences between the two observed spectra (black lines), notably with a more pronounced $f$-peak during Vorcore than Concordiasi. Those are likely due to the different latitudinal sampling during the two campaigns, with more variable $f$ for Vorcore trajectories than Concordiasi's (see the shaded grey area in Fig. 7). However, the main differences between the two campaigns does not lie in the observed spectra but rather in the analyzed ones. Whereas the reanalysis $E_{k_h}$ spectra during Vorcore largely underestimate $E_{k_h}$ and in particular the magnitude of the spectral peak near $f$, there is a clearer spectral peak in the reanalyses with assimilated balloon data. This better performance of the reanalyses during Concordiasi shows that besides adjusting the state of the flow, assimilation of the balloon dataset statistically reinforces some dynamical ingredients in the simulation (here inertial waves).

### 4.2  Impact of underlying model version and resolution on GW representation

#### 4.2.1  Time sampling

Proper representation of the GW variability requires sufficient time resolution, in particular to avoid aliasing of the spectral gap and the $f$-peak. As mentioned above, this is not as much of a critical issue for the model time step (which is small enough not to filter out the GW spatially resolved) as for its time sampling. The time sampling of the reanalysis should indeed be high enough to isolate unequivocally the highest ground-based frequency present in the simulation. Furthermore, this parameter





is one of the most sensitive in trajectory calculations (e.g. Pisso et al., 2010; Bowman et al., 2013), at least with the spatial resolution and integration time step we used. This is especially the case since the winds are instantaneous fields rather than time averages, which implies aliasing when subsampling (e.g. Stohl et al., 1995; Bowman et al., 2013). In order to test the impact of this key parameter on our Lagrangian spectrum estimation, we performed trajectory calculations with ERA5 varying the

time sampling of the reanalysis outputs between 6, 3 and 1 hour. We use the instantaneous ERA5 wind values and do not apply any time averaging which would reduce aliasing but cannot be applied to products with coarser temporal sampling (Hoffmann et al., 2019).

The results of this experiment are presented in Appendix A and can be summarized as follows. First, improving the time sampling of the reanalysis from every 6 to every 3 hours induces large changes in the Lagrangian GW spectrum over the pole

and enables inertial oscillations at periods of $\sim 12$ hours to be resolved. On the contrary, a further improvement of the time resolution from every 3 hours to every 1 hour has little noticeable effect, suggesting that a large fraction of the *resolved* high-intrinsic frequency GW activity is actually caused by the background wind moving the air parcels across "quasi-stationary" wave features with ground-based periods larger than about 6 hours.

### 4.2.2 Impact of vertical and horizontal resolution

The nominal resolution of the model, given in Table 2, largely controls the magnitude of GW fluctuations in reanalyses. This is shown in Figure 8, which displays the fraction of resolved variance in the 2-48 cy/day intrinsic frequency range in the reanalysis products compared to balloon observations, for the horizontal kinetic energy $E_{k_h}$, the potential energy $E_p$, the zonal pseudo-momentum flux $\left(1 - f^2/\hat{\omega}^2\right)|u'w'|$ and the vertical kinetic energy $E_{k_v}$. Considering the (re)analyses from ECMWF (i.e. ERA-Interim, ERA5 and op. ECMWF), the sorting of the resolved variance reflects the horizontal resolution of the products,

for all considered variables. MERRA-2 stands out in this consideration: although its nominal resolution($\sim 60$ km, see Table 2) is better than that of ERA-Interim ($\sim 79$ km). However, this reanalysis has a non spectral the grid, which likely implies some additional diffusion to ensure numerical stability, thus reducing the effective resolution of the model (Holt et al., 2016).

Although the dependency on the reanalysis horizontal resolution is present for all variables considered, it is the more prevalent the lowest the intrinsic frequency (i.e. for the variables on the left in Fig. 8). Indeed, while $E_{k_h}$ than for variables with

variance primarily contained at large $\hat{\omega}$. This is expected when acknowledging that $\hat{\omega}$ is loosely related to the horizontal wavenumber: while $E_{k_h}$ and $E_p$ are already partly resolved in low resolution products and do not depend as much on resolution, $E_{k_v}$ increases strongly when including the additional small-scale brought by the high resolution. Along the same line of argumentation, we note that the fraction of resolved variance increases for the variables depending on low-frequency waves ($E_{k_h}$ and $E_p$) compared to those whose variance is focused at high frequency. In particular, the fraction of resolved $E_p$ is

closer to the fraction of resolved $E_{k_h}$ in the tropics than over the pole where there is a relative increase in $E_{k_h}$ variance at low frequency which is better resolved than the variability at higher frequency. The difference seen between the tropical and polar region in terms of resolved $E_{k_h}$ and $E_p$ variance is almost absent for $E_{k_v}$.

Since GW occur at the mesoscale where the grey zone of global models currently lies, the strong sensitivity to horizontal resolution is expected: models with higher resolution are able to resolved a larger fraction of the spectrum due to their higher





cut-off waevnumber. For a fair comparison of the datasets with different resolutions, we also considered a version of ERA5 truncated in spectral space to match the resolution of ERA-Interim (truncature T255). The GW present are those that can be represented on the ERA-Interim grid; differences hence arise from the different propagation properties and . The corresponding map of vertical wind standard deviation $\sigma_w = \sqrt{2\,E_{k_v}}$ is shown in Fig. 5, panels d). Two interesting results emerge from this

exercise. First, the confinement of regions of high GW activity is similar between close-to-native resolution ERA5 and its spectrally-truncated version, but much less pronounced in ERA-Interim where regions of large GW activity are more spread around. Second, although they effectively have the same resolution, the truncated ERA5 has increased variance compared to ERA-Interim.

Besides horizontal resolution, vertical resolution and vertical mixing formulation have also been shown to play a dominant

role in controlling the wind energetics in atmospheric models (Skamarock et al., 2019). In particular, in their global simulations, Skamarock et al. (2019) found that both the horizontal kinetic energy spectrum and the wind shear (quantified by the gradient Richardson number) largely increased in amplitude with vertical resolution, with the first signs of convergence observed with vertical mesh spacing near 100 m. Near-inertial waves with reduced vertical scales are believed to largely contribute to the increased wind variance at high vertical resolutions (Waite and Snyder, 2009; Skamarock et al., 2019). The dispersion relation:

$$m^2 = \frac{N^2 - \hat{\omega}^2}{\hat{\omega}^2 - f^2}(k^2 + l^2) \tag{11}$$

indeed shows that when $\hat{\omega} \to f$, the vertical wavelength is reduced, and these waves are potentially more sensitive to the vertical resolution. In accord with those previous works, our analysis suggests a specific difficulty to simulate small vertical wavelength waves near the inertial frequency. The $E_{k_h}$ amplitude of GWs of frequency between 2 and 4 cy/day are only slightly

underestimated in the tropics (see Fig. 3) where they have larger vertical scales ($f \to 0$ in Eq. 11), whereas it is much reduced in the polar latitude (Fig. 3) where those waves are near inertial and hence close to critical levels. However, the different reanalysis products from the ECMWF do not exhibit large changes in the description of the f-peak although the vertical resolution of the model in the lower stratosphere varies by more than a factor of four between ERA-Interim and ERA5 (Table 2). This is the case whether or not balloon data are assimilated (Fig. 7). It is likely that the limited vertical resolution of the reanalyses in the

lower stratosphere is too low or the vertical mixing too large for the sensitivity to resolution to emerge.

### 4.3 Implications for Gravity Waves studies and Lagrangian modeling based on reanalyses

A simple recommendation can be drawn from the analysis presented above. As pointed out by a number of studies, gravity waves are now partly represented in modern reanalysis, in particular if one considers the temperature and horizontal wind fluctuations, which are tied to the low frequency part of the gravity wave intrinsic frequency spectrum. In that context of low

frequency-large scale waves, the use of reanalysis to quantify atmospheric gravity wave variability is justified, in particular for case studies in the tropics. However, the fraction of the variability resolved by the reanalyses gets smaller with the scale of the waves that generate that variability, so that the momentum flux (intermediate scale waves) and a fortiori the vertical wind variability (tied to small-scale waves) are still underestimated.





# 5 Conclusions

This paper examined the representation of gravity wave induced fluctuations in the lower stratosphere of modern reanalyses through the computation of Lagrangian trajectories and their comparison with quasi-Lagrangian balloon observations. Consistent with previous studies, we find that reanalysis systems resolve a significant part of the gravity wave spectrum at low
intrinsic frequencies, and represent part of the associated variability in temperature and horizontal wind. In particular, specific characteristics of the observed spectrum can be well reproduced, such as the spectral gap between gravity waves and the large-scale flow, or the increased horizontal wind variance near the Coriolis frequency. However, the resolved fraction of the variability decreases with increasing intrinsic frequency so that the vertical wind variability, tied to high intrinsic frequency, is largely underestimated in modern systems. The momentum flux, peaking at intermediate scales, is also underestimated, though
less than vertical velocity.

Generally, the analyses examined fulfill the expectation that the higher the nominal horizontal resolution, the larger the fraction of resolved variability. Contrasting the tropics with the southern high latitudes further suggests that small-vertical wavelength waves (approaching the inertial frequency) at high latitudes are not well represented. Despite the capability of the underlying models, we also found that the waves in the analyses are not purely self generated by the models, but may rely for
a significant part on data assimilation. In particular we found that the assimilation of long-duration balloon observations over Antarctica in austral spring enhanced GW activity in the reanalyses. In any case, the fact that GW are present in NWP models suggest that observations with GW signature (radiosondes, GPS radio-occultations) could be successfully assimilated.

In the future, improved vertical and horizontal resolutions (and maybe specific data assimilation strategies) will allow NWP models to resolve an increased fraction of GW induced variability, which will make them even more valuable than currently
for GW related studies. The increased presence of this high-frequency horizontal wind variability requires using higher time sampling (3 hourly) for accurate trajectory calculations. However, we expect that vertical wind variability, which is tied to the smallest spatial scales ($\sim$1-10 km) and for which current modeling approximations (such as the hydrostatic one) are not valid will still remain a challenge. This emphasizes the long-term need for gravity wave parameterizations in NWP-based process studies, especially for processes depending on vertical velocity.

# 25 Appendix A: Aliasing in the frequency spectrum due to the finite output frequency of the reanalyses

As indicated by Eq. 5, to a given intrinsic frequency corresponds a range of horizontal wavenumbers and ground relative frequency. It is unclear what the contribution of low and high horizontal phase speed is, and whether part of the missing activity





results from undersampling the time space. Indeed, the horizontal kinetic energy (for example) at a given intrinsic frequency $E_{k_h}(\hat{\omega})$ actually depends on different ground-based frequencies:

$$E_{k_h}(\hat{\omega}) = \int\limits_{-\infty}^{+\infty}\int\limits_{-\infty}^{+\infty}\int\limits_{-\infty}^{+\infty} \delta\left(\omega = \hat{\omega} + k\bar{u} + l\bar{v}\right) E_{k_h}(\omega, k, l)\mathrm{d}\omega\mathrm{d}k\mathrm{d}l \qquad (A1)$$

where $\delta$ is the 1D Dirac distribution and $E$ is the 3D-power spectral density (note that only 3D integration is necessary since for

internal gravity waves the dispersion relation directly imposes the vertical wavenumber given the three other wave parameters and the background). It is apparent that a given $\hat{\omega}$ corresponds to a whole surface of $\omega$ and wavenumbers. High intrinsic frequency waves may hence have high ground relative frequency or be short-scale waves in a large background wind. The two cases are affected differently by our analysis method. Since we retrieve the reanalysis at horizontal resolution close to the actual grid spacing, we fully resolve analysis waves which have high intrinsic frequencies due to their small horizontal scales

(large $k\bar{u}$). However, the reanalyses are sampled at a time resolution of typically 1 to 6 hours, i.e. lower than the model time step (less than 15 minutes). There can be aliasing in the $\hat{\omega}$ for high frequncy waves with large scales but high $\omega$. In order to evaluate this effect, we take advantage of the 1-hourly ERA5 to compare trajectories using 1, 3 or 6 hourly reanalysis fields.

The results are displayed in Fig. A1 for Vorcore flights. Large differences occur for both the $E_{k_h}$ spectrum and the rotary ratio between 6 and 3 hours: the inertial peak at $\hat{\omega} \simeq 12$ hours is completely aliased with the resolution of 6 hours (for which

it is close to the Nyquist frequency), but resolved at 3 hours. This shows that the waves dominating the energy spectrum at frequencies near $f$ have predominantly large scales and $\hat{\omega} \simeq \omega$. On the contrary, further improving does not affect much the $E_{k_h}$ spectrum, but leads to a more realistic rotary ratio at higher frequencies. This exercise demonstrates that the 3-hourly resolution provided by most reanalyses (except JRA55) is sufficient for our purpose, i.e. evaluating the GW intrinsic-frequency spectrum.

**Appendix B: Comparison of GW Lagrangian spectra estimates using isopycnic, isentropic, kinematic and diabatic trajectories**

To test the validity of the assumptions made in Sect. 2.1 to use balloon isopycnic trajectories to infer Lagrangian spectra, we computed isopycnic, isentropic and 3D-kinematic and 3D-diabatic trajectories in the reanalyses. The resulting $E_{k_h}$ and $E_p$ spectra are presented in Fig. B1 for ERA5 with two time resolutions: 3 and 1 hourly. For the $E_{k_h}$ spectrum, the 3 calculations

converge at 1 and 3 hourly. For the $E_p$ spectrum, only the kinematic case with 3 hourly winds disagrees with the others. The discrepancy is resolved, suggesting aliasing of high frequency vertical motions at 3 hourly resolution ; this effect is filtered out in the isopycnic and isentropic cases, for which the vertical wind is implicitly accounted for through the fully resolved displacement of the $\theta$ surfaces.



*Author contributions.* AP designed the study and carried out the analysis. AH and RP contributed to the analysis. AP and AH developed the balloon trajectory code used in the study. BL guided the use of reanalysis data. AP wrote the paper, with contributions from all co-authors.

*Competing interests.* The authors declare that they have no conflict of interest.

*Acknowledgements.* AP thanks Dr Manfred Ern for his helpful comments on an earlier version of the manuscript. ERA5 results were
5    generated using Copernicus Climate Change Service Information 2019.





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





**Figure 3.** Intrinsic frequency spectra of (top) horizontal kinetic energy per unit mass $E_{k_h}$ (solid) and rotary ratio $R(\hat{\omega})$ (dashed), (middle) potential energy per unit mass $E_p$ and ratio of kinetic over potential energy $E_{k_h}/E_p$, and (bottom) zonal pseudo-momentum flux per unit mass $\left(1 - f^2/\hat{\omega}^2\right)|\overline{u'w'}|$ during the polar Vorcore campaign (left column) and the equatorial PreConcordiasi (right column) campaign. The black curve on each plot corresponds to balloon observations while the spectra estimated from isopycnic trajectories in different (re)analyses are in colors. The trajectories used to compute the spectra are 8-days trajectories started at the balloon position.





**Figure 4.** Probability density functions (PDF) of (top) temperature fluctuations, (middle) zonal momentum flux and (bottom) vertical wind in the different reanalyses (black) and in the balloon observations (colors), for intrinsic frequencies between $f$ and 48 cy/day (left, pole) or between 1 and 48 cy/day (right, tropics).



**Figure 5.** Global maps of vertical velocity standard deviation $\sigma_w$ at 100 hPa in different reanalyses systems during March 2010. Note that the color scales have been adjusted so that the ratio between local and global $\sigma_w$ corresponds to the same color in all panels. ERA5-lr is a retrieval from ERA5 reanalysis truncated at the same resolution as the ERA-Interim (T255).



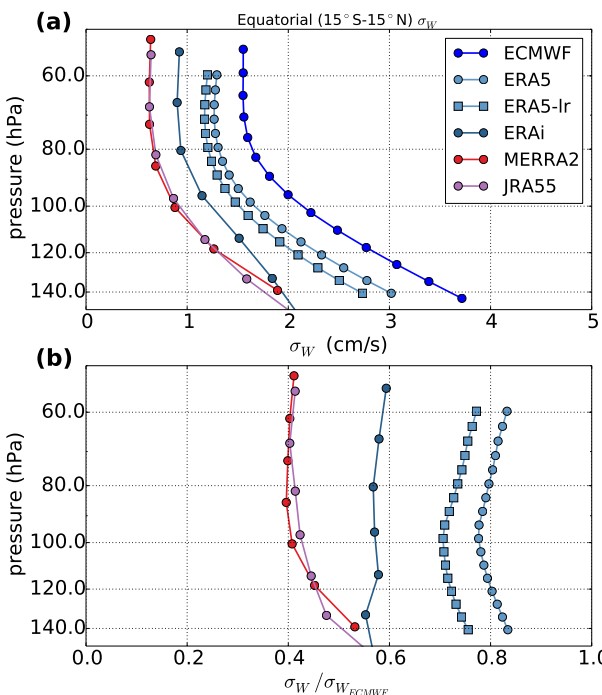

**Figure 6.** (Top) Equatorial (15°S-15°N) average profile of vertical velocity variance in different reanalyses systems for March 2010. Bottom: the same, but normalized by the profile from the 2010 ECMWF operational analysis. ERA5-lr is a retrieval from ERA5 reanalysis truncated at the same resolution as the ERA-Interim (T255). The position of the dots corresponds to the model vertical levels, and emphasize the better resolution of ERA5 in the Tropical Tropopause Layer compared to earlier products.





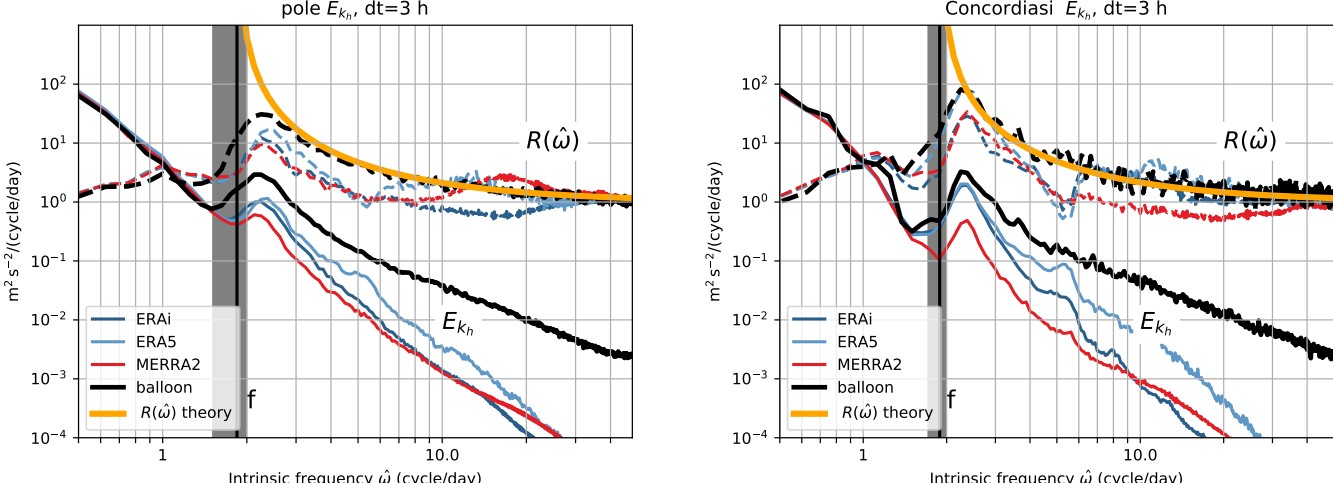

**Figure 7.** Intrinsic frequency spectra of horizontal kinetic energy per unit mass $E_{k_h}$ during the Vorcore (austral winter 2005, left) and Concordiasi (austral winter 2010, right) campaigns. The black curve corresponds to balloon observations during the campaigns while the spectra estimated from isopycnic trajectories in different (re)analyses are in colors. The vertical black line indicates the mean Coriolis parameter $f$ while the shaded grey area represents its variability along the balloon trajectories.



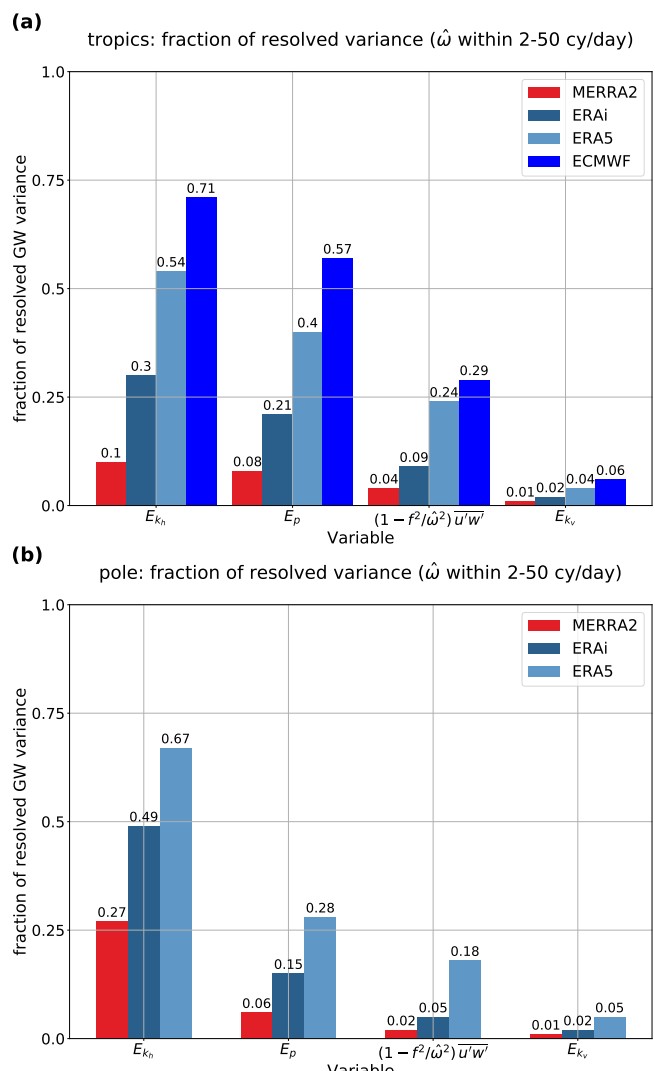

**Figure 8.** Bar chart of the fraction of resolved variance in the reanalyses balloon trajectories in the 2-48 cy/day frequency range, normalized by the variance present in balloon observations. Results are shown for the (a) tropical 2010 PreConcordiasi campaign and the (b) 2005 polar latitudes'Vorcore. The fields considered are, from left to right: horizontal kinetic energy $E_{k_h}$, potential energy $E_p$, zonal pseudo-momentum flux $\left(1 - f^2/\hat{\omega}^2\right)|\overline{u'w'}|$ and vertical kinetic energy $E_{k_v}$.



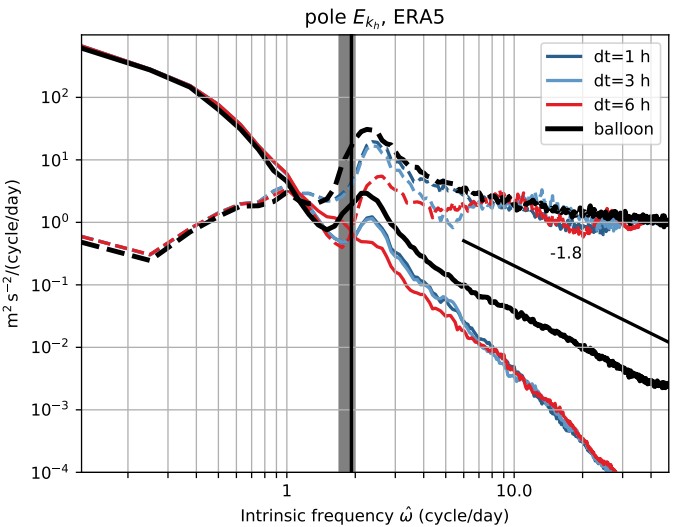

**Figure A1.** Intrinsic-frequency spectrum of kinetic energy per unit mass ($E_{k_h}$, solid lines) and rotary ratio (dashed lines) along ERA5 trajectories started at Vorcore balloon locations, for different reanalysis output frequencies: 1, 3 and 6 hours.



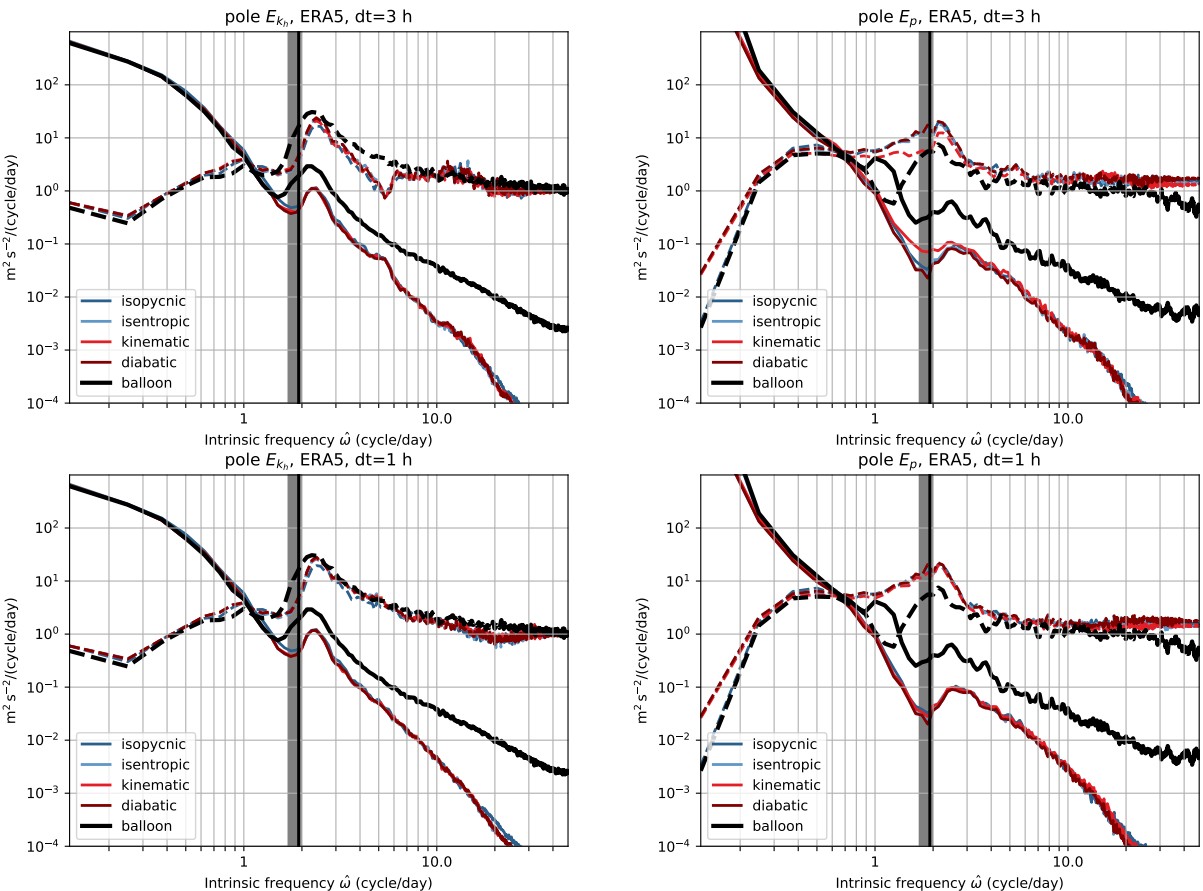

**Figure B1.** Intrinsic-frequency spectrum of (left) kinetic energy $E_{k_h}$ and (right) potential energy $E_p$ per unit mass along ERA5 trajectories started at Vorcore balloon locations, for different trajectory types: isopycnic (constant $\rho$; note that $E_p$ spectrum is in that case adjusted following Eq. 3, as is done for the balloons), isentropic (constant $\theta$) and kinematic and different reanalysis output frequencies (top: 3 hours; bottom: 1 hour).