# Peer review of "Lagrangian Gravity Wave spectra in the lower stratosphere of current (re)analyses"

_Atmospheric Chemistry and Physics, 2020_

## Referee Comment (RC1) · Corwin Wright (Referee) · 8 Apr 2020

Corwin Wright (Referee)

c.wright@bath.ac.uk

The authors use data from long-duration quasi-Lagrangian super-pressure balloons (SPBs) to assess the accuracy of the GW spectrum simulated in the ERA-Interim, ERA5, and MERRA-2 reanalyses.

They use data from three campaigns, one of which was assimilated into the reanalyses, with pressure and temperature measured directly by SPB instrumentation and wind speed inferred from GPS position data. The particular advantage of using these SPB data is that, unlike ground- or space-based measurements, they record data in a Lagrangian frame of reference.

Time sampling of these data may be a factor in their results, but this is taken into ap-

propriate consideration throughout the paper. In addition, the authors describe a wide range of other possible error sources - the discussion of these is often slightly briefer than I would prefer, but demonstrates that they have been taken into consideration.

The reanalyses are shown to reproduce important features of the wave spectrum, including the PW-GW spectral gap and a peak near the inertial frequency at high latitudes. The authors conclude that the reanalyses do a good job of reproducing GW features in the observed data, particularly at low frequency, but that the high-frequency performance is sufficiently deficient that GW parameterisations will still be required in the medium to long term. They also note that the assimilation of the balloons for one campaign had a positive impact on model quality.

The paper is very clearly written, in particular with a very high standard of written scientific English, and I see no critical scientific deficiencies. I have listed a few minor issues below, but none of these are critical, and I would support publication with at most minor changes.

1. The authors mention JRA-55 a couple of times early on, but then rapidly remove it from consideration due to time-sampling issues. However, I don't think they actually use these data anywhere significant in the paper. For clarity I think it would be best to just remove JRA-55 and mentions thereof from the paper completely. This is particularly a problem for the abstract, as it is potentially misleading for someone looking for an assessment of this model specifically.

2. Figure 2b makes the pressure-level differences look bigger than they actually are, so might be worth mentioning that the y-scale is over a narrow range (roughly 0.5km max deviation - for ERA5, which is the highest vertical resolution of those considered, this is only ~2-3 model levels at these heights)

3. Figures 3 and 7: the panels on the right-hand side are labelled "PreConcordiasi" but those on the left do not say "VorCore", but "pole" instead. I would suggest labelling the left panels as Vorcore to make it immediately clear.

4. Figure 3: it is quite hard to see the relationship between the values in right-hand panels due to the thickness of the black line and how much it jumps around on top of the red and blue lines. I would suggest replotting it somehow so that the reader can actually see the coloured lines - maybe make the black lines thinner and reduce the heaviness of the gridlines to compensate visually?

5. P12L31: is this specifically zonal momentum flux?

6. Figure 5 uses a jet colour table. This is hard for colourblind readers to read, and also suggests semantic meaning at sharp colour transitions where none is implied by the data. I would strongly suggest changing the colour table used for this figure. Also, some of the maximal regions are out of band on the colour table and plotted in white - it may be useful to truncate the data at these points to avoid this issue.

Typos/Grammar:

P06L11 non-grammatical: Missing "which"?

P12L15 non-grammatical: "provide information on"?

P14L13 non-grammatical: "since Concordiasi only ran/flew for one year", maybe?

P15L21 non-grammatical: "has a non-spectral grid"?

P15L24 non-grammatical: "more prevalent the lower the"

P16L01 typo :"wavenumber"

---

## Referee Comment (RC2) · Anonymous Referee #2 · 7 May 2020

The authors use balloon data to assess the gravity wave spectrum in various reanalyses and one operational analysis. Although they find that that reanalyses represent the shape of the spectrum well, the variability is lacking compared to the balloons especially at higher intrinsic frequencies. Models with higher horizontal and vertical resolution represent the gravity wave variability better, although vertical resolution seems to have less of an effect than might be expected. They also show that including the balloon observations in the reanalyses improves the representation of gravity wave variance at low frequencies.

This paper is very well written and clearly organized. The results are very relevant and of great interest to modelers. These results will help give guidance to modelers trying to improve the representation of gravity waves in their models, both explicit and parameterized waves. I recommend this paper be published with a few minor considerations below.

p. 5, line 24: "Furthermore, due to their expected small horizontal scale the importance of non hydrostatic effects..." Should there be an "and" in here? Otherwise this sentence doesn't really make sense to me.

p.10, line 19: According to this equation,  $R(\omega)$  should go to 0 as  $\omega$  approaches f, but the Figure shows  $R(\omega)$  goes to infinity as  $\omega$  approaches f.

p. 12, line 21: I would say "The latter behavior..." instead of "This last behavior..."

p. 15, line 29: I would say "..., it is more prevalent at the lowest intrinsic frequencies..." also, pronounced would be a better word than prevalent.

p. 15, lines 29-34: What about the influence of vertical resolution on this plot? In particular it seems like there is a clear distinction between the higher vertical resolution models and lower vertical resolution models in the u'w' columns for both pole and tropics.

p. 15, line 30: This sentence doesn't really make sense grammatically: "Indeed, while Ekh than for variables with variance primarily contained at large w." I suggest maybe "Indeed, the dependency on horizontal resolution is more pronounced for Ekh than for variables with variance primarily at large w."

p. 16, lines 6-14: What about adding the truncated ERA5 to Figure 8? Would this provide more clues to the importance of horizontal vs vertical resolution?

p. 16, line 10: broken off sentence: "... arise from the different propagation properties and ."

p. 25, Figure 4: The labels are quite tiny.

2020.

---

## Author Comment (AC1) · 9 Jun 2020

**Reply to Corwin Wright**

We would like to thank Corwin Wright for his careful review and insightful comments on our paper. Please find below our point by point reply.

1. *Reviewer* — The paper is very clearly written, in particular with a very high standard of written scientific English, and I see no critical scientific deficiencies. I have listed a few minor issues below, but none of these are critical, and I would support publication with at most minor changes. 1. The authors mention JRA-55 a couple of times early on, but then rapidly remove it from consideration due to time-sampling issues. However, I don't think they actually use these data any-

[Figure]

where significant in the paper. For clarity I think it would be best to just remove JRA-55 and mentions thereof from the paper completely. This is particularly a problem for the abstract, as it is potentially misleading for someone looking for an assessment of this model specifically.

*Authors* — We agree that it is misleading to mention JRA-55 in the abstract, because we were unable to evaluate its intrinsic frequency spectrum. However, we find it worth showing that the general behavior of this renalaysis is similar to the others in terms of spatial variability (Figures 5 and 6), and also that the Lagrangian approach of GW evaluation cannot be applied to that product due to its coarse time sampling. Hence, we removed mentions of JRA-55 in the abstract but chose to keep them in the main body of the paper.

2. *Reviewer* — Figure 2b makes the pressure-level differences look bigger than they actually are, so might be worth mentioning that the y-scale is over a narrow range (roughly 0.5km max deviation - for ERA5, which is the highest vertical resolution of those considered, this is only ∼2-3 model levels at these heights)

   *Authors* — Now mentioned

3. *Reviewer* — Figures 3 and 7: the panels on the right-hand side are labelled "PreConcordiasi" but those on the left do not say "VorCore", but "pole" instead. I would suggest labelling the left panels as Vorcore to make it immediately clear.

   *Authors* — Thank you, we agree. The figures have been changed as suggested.

4. *Reviewer* — 4. Figure 3: it is quite hard to see the relationship between the values in right-hand panels due to the thickness of the black line and how much it jumps around on top of the red and blue lines. I would suggest replotting it somehow so that the reader can actually see the coloured lines - maybe make the black lines thinner and reduce the heaviness of the gridlines to compensate visually?

*Authors* — Thank you for the suggestion, we have adjusted Figure 3, which is more legible now.

5. *Reviewer* — 5. P12L31: is this specifically zonal momentum flux?

    *Authors* — Yes, this is now specified.

6. *Reviewer* — Figure 5 uses a jet colour table. This is hard for colourblind readers to read, and also suggests semantic meaning at sharp colour transitions where none is implied by the data. I would strongly suggest changing the colour table used for this figure. Also, some of the maximal regions are out of band on the colour table and plotted in white - it may be useful to truncate the data at these points to avoid this issue.

    *Authors* — We were not aware of this, thank you for raising that point. The colormap has been changed for Viridis.

    *Authors* — Typos and grammar mistakes have been corrected, thank you for pointing them out.

---

## Author Comment (AC2) · 9 Jun 2020

**Reply to reviewer 2**

We thank reviewer 2 for their evaluation of our paper and the constructive comments. Their suggestions for improvement have been taken into account. Please find below our point by point reply.

1. ***Reviewer*** — The authors use balloon data to assess the gravity wave spectrum in various reanalyses and one operational analysis. Although they find that that reanalyses represent the shape of the spectrum well, the variability is lacking compared to the balloons especially at higher intrinsic frequencies. Models with higher horizontal and vertical resolution represent the gravity wave variability bet-

[Figure]

ter, although vertical resolution seems to have less of an effect than might be expected. They also show that including the balloon observations in the reanalyses improves the representation of gravity wave variance at low frequencies.

This paper is very well written and clearly organized. The results are very relevant and of great interest to modelers. These results will help give guidance to modelers trying to improve the representation of gravity waves in their models, both explicit and paramereterized waves. I recommend this paper be published with a few minor considerations below.

*Authors* — Thank you

2. *Reviewer* — "Furthermore, due to their expected small horizontal scale the impor- tance of non hydrostatic effects..." Should there be an "and" in here? Otherwise this sentence doesn't really make sense to me.

*Authors* — Yes, thank you. This has been corrected.

3. *Reviewer* — p.10, line 19: According to this equation, R($\omega$) should go to 0 as $\omega$ approaches f, but the Figure shows R($\omega$) goes to infinity as $\omega$ approaches f.

*Authors* — Actually, there was a sign error in the figures: the black line was depicting $|f| = -f$ (since Vorcore flew in in the Southern hemisphere) instead of $f$. Thank you for pointing this out, it is now corrected. We also specifically warn the reader about the sign of $f$ below the formula.

4. *Reviewer* — p. 12, line 21: I would say "The latter behavior. . ." instead of "This last behavior. . ."

*Authors* — Changed as suggested.

5. *Reviewer* — p. 15, line 29: I would say "... , it is more prevalent at the lowest intrinsic frequencies..." also, pronounced would be a better word than prevalent.

*Authors* — Changed as suggested.

6. *Reviewer* — p. 15, lines 29-34: What about the influence of vertical resolution on this plot? In particular it seems like there is a clear distinction between the higher vertical resolution models and lower vertical resolution models in the u'w' columns for both pole and tropics.

   *Authors* — Not exactly, since ERA5 has a higher vertical resolution than ECMWF(see Table 2). This point is now mentioned in the manuscript.

7. *Reviewer* — p. 15, line 30: This sentence doesn't really make sense grammatically: "Indeed, while Ekh than for variables with variance primarily contained at large w." I suggest maybe "Indeed, the dependency on horizontal resolution is more pronounced for Ekh than for variables with variance primarily at large w."

   *Authors* — Changed as suggested

8. *Reviewer* — p. 16, lines 6-14: What about adding the truncated ERA5 to Figure 8? Would this provide more clues to the importance of horizontal vs vertical resolution?

   *Authors* — Comparing the degraded ERA5 to the full ERA5 only provides us with a lower boundary for the dependence on horizontal resolution. Indeed, although it filters out small-scale waves, the low-resolution ERA5 has the same (high-resolution) sources of large-scale waves as the full ERA5, so that it still contains "information" provided by the high horizontal resolution for that part of the spectrum. Because of that, ERA5-lr shound not be considered a high vertical resolution ERA interim, and the difference between ERAi, ERA5-lr and ERA5 cannot be solely attributed to either horizontal or vertical resolution. We now explain this point in more details in the manuscript.

9. ***Reviewer*** — p. 16, line 10: broken off sentence: ". . . arise from the different propagation properties and ."

   ***Authors*** — Missing "Sources". This has been corrected.

10. ***Reviewer*** — p. 25, Figure 4: The labels are quite tiny.

    ***Authors*** — We agree. They have been enlarged.